# Mateo Tepe or Devils Tower: Native and Tourist Differences in Geosite Interpretations

Richard Stoffle [1,*], Kathleen Van Vlack [2], Heather H. Lim [1] and Alannah Bell [1]

1   School of Anthropology, University of Arizona, Tucson, AZ 85721, USA; hl348@cornell.edu (H.H.L.); ajb6uwc@virginia.edu (A.B.)
2   Applied Indigenous Studies, Northern Arizona University, Flagstaff, AZ 86011, USA; kathleen.van-vlack@nau.edu
*   Correspondence: rstoffle@arizona.edu; Tel.: +1-(520)-907-2330

**Abstract:** Devils Tower is located in Wyoming, USA. It is composed of volcanic elements from the Tertiary Period of geologic time. Geologists are interested in this geosite because it is a unique, upstanding, steep-sided, high-relief exhumed Tertiary-age volcanic plug. As a Native American cultural geosite, however, it is often called *Mateo Tepe*, and it is a sacred place to over 20 Native American tribes. It was inscribed as America's first national monument in 1906 by President Theodore Roosevelt, because of its special geology. It is visually dramatic due to its columns, which are understood by earth scientists as a wonder of geology but by Native people as the claw scratches of a spiritual bear. These vertical cracks are the focus of rock climbers and Native people, respectively as opportunities for adventure and self-fulfillment and spiritual paths to another dimension and the achievement of religious balance in the world. Mateo Tepe became a national monument due to it being a unique geologic feature. The geopark concept is used in this analysis to talk about this geologically based monument.

**Keywords:** Devils Tower; sacred geography; sacred space; Native Americans; geosites; geoheritage; geopark; National Park Service; pilgrimage; Mateo Tepe





## 1. Introduction

> *When I was a boy, I often accompanied my father on trips and he would point out various features of the landscape and tell me the name and stories associated with them... imprinted on my mind was the fact that the Sioux people cherished their lands and treated them as if they were people who shared a common history with humans* —Vine Deloria Jr. [1].

After tens of thousands of years of serving as a key living component of a sacred geographical landscape in the Black Hills, Mateo Tepe (Bear Lodge) was taken from Native people and reconceptualized as an inert volcanic butte, the purpose of which is argued to be best reserved for tourists. Scholars have documented that the Black Hills are a component of a living sacred geographic landscape, which is a Mirror of Heaven, and Mateo Tepe is the Staircase to the Stars [2–5].

This case of (re)conceptualizing a geological feature in Wyoming, USA in a popular national monument reflects a worldwide rethinking of nature over the past generation. While all natural places are now reviewed for what is called "natural or human services"—that is, their contributions to human society—some natural services are now protected because they are sacred to some people [6–8]. In a book entitled *Mount Sacred: Brief Global History of Holy Mountains Since 1500* [9], Mathieu [9] compiles essays from many cultures about the modern recognition and protection of sacred values associated with mountains and other geological areas. These reconceptualizations often involve renaming the area from its colonial name to its Indigenous name, incorporating Indigenous people into land

management, new interpretation and education, and regulation of touring behaviors to achieve sustainability. Some examples include shifting Ayers Rock to Uluru [10,11] in Australia, shifting the meaning of Mt. Baekdu on the Korean–Chinese boundary [12], and having a president's name on Mount McKinley replaced by Denali in Alaska, USA [13]. These shifts in meaning and terms of reference have been supported by the International Union for Conservation of Nature since 1990 [14].

This is a Native American heritage analysis of a reconception of a geosite based on ethnographic tribal interviews, legal documents, and published literature. Together they establish the spiritual and cultural implications of shifting traditional patterns of interactions as a path to heaven to its national monument management as an inert feature. The human services of Mateo Tepe have thus shifted from religious and ceremonial to economic and recreation.

Today the U.S. National Park Service (NPS) is working to rectify past exclusions by bringing Native American peoples formally into U.S. National Park cultural resource consultations, interpretations, and management. These activities are termed co-stewardship of federal lands and water and are mandated by the Department of the Interior [15].

In 1879, an official geological map of the Black Hills of Dakota was produced by Henry Newton, E.M. Department of the Interior, while serving on the U.S.G. and G. Survey, which was headed by J.W. Powell. Devils Tower was given the official place name of Bear Lodge on this 1879 map. The Lakota place name Mateo Tepe translates as *Bear Lodge* (Figure 1).

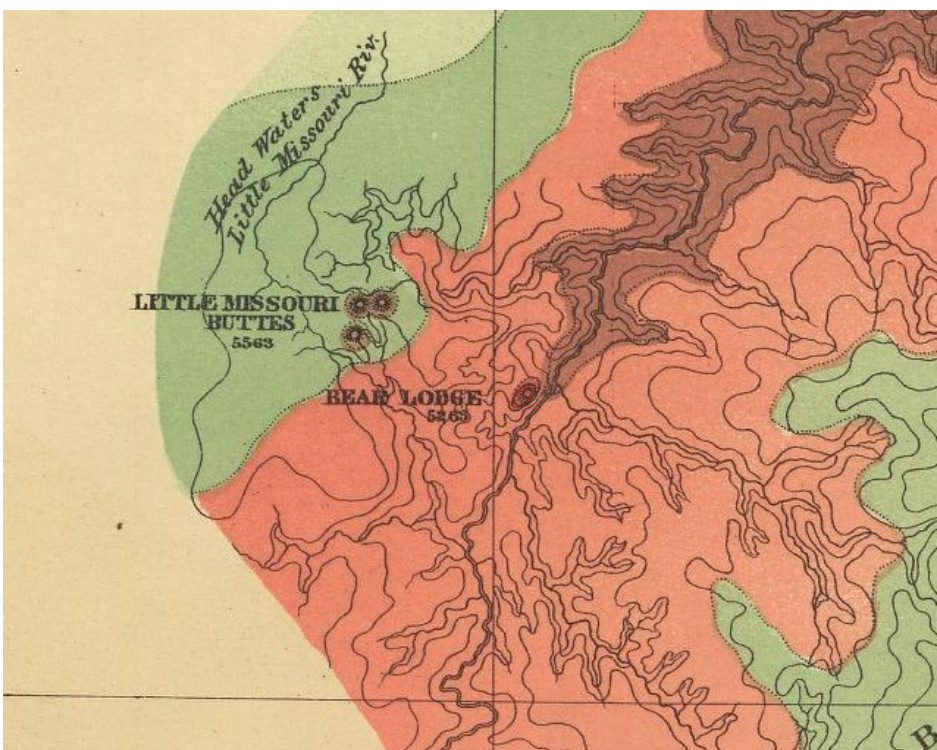

**Figure 1.** 1879 Survey map of Black Hills with Bear Lodge (Mateo Tepe) used to denote what is now called Devils Tower [16]. The different colors represent different hydrological systems.

The worldwide issue of sustainable heritage tourism is addressed in this analysis, especially as it is considered in geoparks and geosites that were initially nominated for their charismatic geology [17–19]. It is argued here that heritage values of cultural groups attached to geologically special places should be considered alongside those of tourists, who focus on the geology of place andregion [20,21]. Cross-cultural environmental communication [22] is a key issue, especially when epistemological divides are involved [23]. It is challenging when managers of geoparks and geosites must be responsive to various layers and types of heritage values [24].

## 2. Sacred Space Background

Sacred space is a key concept for this essay, and thus it is important to situate it in a broader literature and temporal frame. To facilitate this goal, we present key understandings from an analysis of Taos Pueblo's legal arguments to reclaim their sacred cultural landscape surrounding Blue Lake in New Mexico. Hughes and Swan [25] present a summary of the sacred lands concepts that apply to the Taos Blue Lake landscape case, and these are relevant to this cultural analysis of Mateo Tepe.

- Point A: The U.S. Congress acknowledged, as it later did more explicitly in the American Indian Religious Freedom Act, that Native American Indian tribes recognize certain places as sacred space, an attitude which is found in all tribes. The Lakota and others have a spiritual relationship withMateo Tepe (Bear Butte) in the Black Hills, and both the Navajo and Hopi regard the San Francisco Peaks near Flagstaff as sacred.
- Point B: The conception of the earth as sacred is widespread among traditional peoples around the world and through history. The ancient Chinese practice of feng shui (geomancy) treats the landscape as a network of potent spots connected by lines of energy. One would be foolish, its practitioners believed, to ignore this sacred geography when locating a house, road, or temple. The Greek philosopher Plato affirmed that the earth is a living organism, alive in every part, and also that there are particular locations where spiritual powers operate positively or negatively.
- Point C: In the traditional Native American view, all of nature is sacred, but that in certain spots, the spirit-power manifests itself more clearly. It is to those places that a person seeking a vision would make a quest. They are localities where the great events of tribal history and the era of Creation took place. The Native American view of the universe is that of a sacred continuum that contains foci of spirit-power.
- Point D: Places of power often are distinguished by a natural feature: an impressive grove of large old trees, a pure spring, a deep lake, a fissure in the earth, or a mountain peak. These charismatic places are components of sacred landscapes of great natural beauty. Spiritual people take account of the lay of the land and the mountain forms visible from it. Within these landscape boundaries, all human use other than religious worship was forbidden.

Before the Taos Pueblo legal case regarding the expansion of the Taos city public airport flights over Blue Lake, there were a few arguments for the protection of cultural landscapes. T. J. Ferguson, the professor charged with compiling the environmental impact analysis (EIS) of the Blue Lake study area, worked with Taos Pueblo representatives to document and locate dozens of Traditional Cultural Properties (TCP). In order to have an impact on the geographically wide study area, however, the arguments to protect Blue Lake area required that all these TCPs be considered as functionally integrated components of a Taos Pueblo heritage landscape centered over Blue Lake [26,27]. The Taos heritage study successfully made this heritage landscape argument, and the proposal for a new airport runway that would put flights over Blue Lake was refused.

On 15 December 1970, President Richard Nixon signed H.R. 471 Blue Lake Bill Taos-Pueblo American Indian Land Deed (National Archives ID 66394205). President Richard Nixon Spoke at the Signing Ceremony for the Blue Lake Bill, HR 471, which Deeded Lands to Taos Pueblo Indians, (National Archives ID 7268141). President Nixon said:

> *This bill represents justice, because in 1906 an injustice was done in which land involved in this bill was taken from the Taos Pueblo Indians. And now Congress of the US returns that land to whom it belongs. The bill also involves respect for religion. We realize that for 700 years the Taos Pueblo Indians worshiped in this place. We restore this place of worship to them for all the years to come [28].* Now 50 years later, Native cultural landscapes largely remain unconsidered in environmental studies and unprotected as heritage places.

Mateo Tepe has become famous around the world and in North America both due to its charismatic geology and an ongoing dispute regarding what it means culturally to Native

Americans, to the worldwide climbing community, and to tourist visitors (Figures 2 and 3). Native American people understand the tall polygonal columns as paths to another dimension that were made by a spiritual bear. Mountain and rock climbers ascend the cracks and rejoice in their climbing victories. This paper analyzes the dispute over the meaning and potential management priorities of this geosite and the future of this geopark.

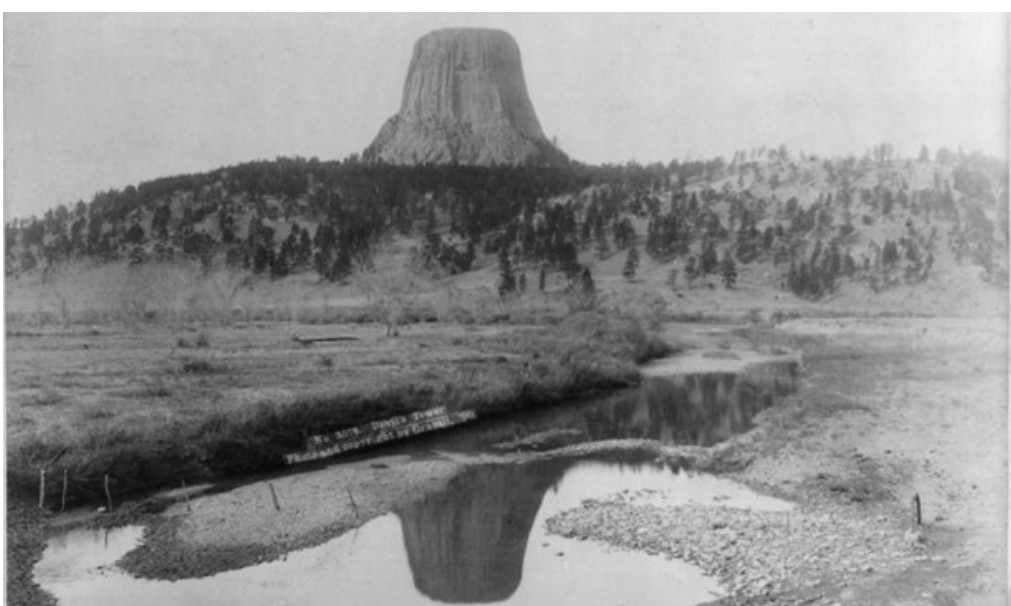

**Figure 2.** Historic photo of Devils Tower and Belle Fourche River [29].

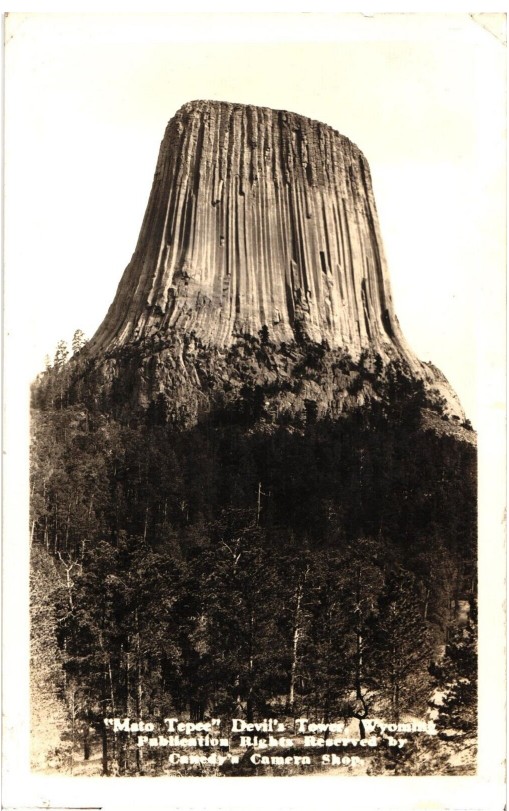

**Figure 3.** 1930s Tourist post card of Mateo Tepe or Devils Tower [30].

Devils Tower is a high-relief igneous rock mega-columnar formation rising some 386m above the wide flat valley of the nearby Belle Fourche River (Figure 2). It is comprised of

porphyritic phonolite and is the world's largest example of polygonal columnar jointing [31]. On its lower slopes, the towering butte is flanked by an apron of talus deposits. There are a number of theories as to its geologic origin, including a volcanic plug, laccolith deposits, and a maar-diatreme origin [32]. At present, geologists prefer the theory that it is an exhumed volcanic neck that intruded into the sedimentary rock sequences above it about 40.5 Ma ago [33]. Geomorphically, it formed as an exhumed resistant rock body as the enclosing and overlying softer Mesozoic sedimentary rock formations of sandstone, shale, and gypsum were eroded away.

From a Native American perspective, Mateo Tepe is an iconic geological feature and thus an example of what some anthropologists call a self-voiced protuberance. Dramatically rising from a flat high plains landscape containing a major river, from a human perspective, it is the most visible and important component of the topography for hundreds of square miles. This conclusion is attested to by the various culturally derived social constructions of its formation, meaning, and subsequent appropriate management recommendations [4,5,34]. Sundstrom [3] argues that the sacred sites of the Black Hills are perceived by Native American people as reflecting the pattern of falling stars. After 1890, Amos Bad Heart Bull drew a sacred geography map of the Black Hills region including Mateo Tepe. Nickolas Black Elk, a well-known Oglala Lakota holy person, recorded a series of the ancient Falling Star accounts in the 1930s and 1940s. In these Lakota accounts, Falling Star is a hero who travels through the Black Hills, has adventures, and visits the seven Star Villages, which are associated with the Pleiades or the Big Dipper. These villages occur both on Earth in the Black Hills and in the Heavens [3] (Figure 4). Places and events in the two dimensions are mirrored.

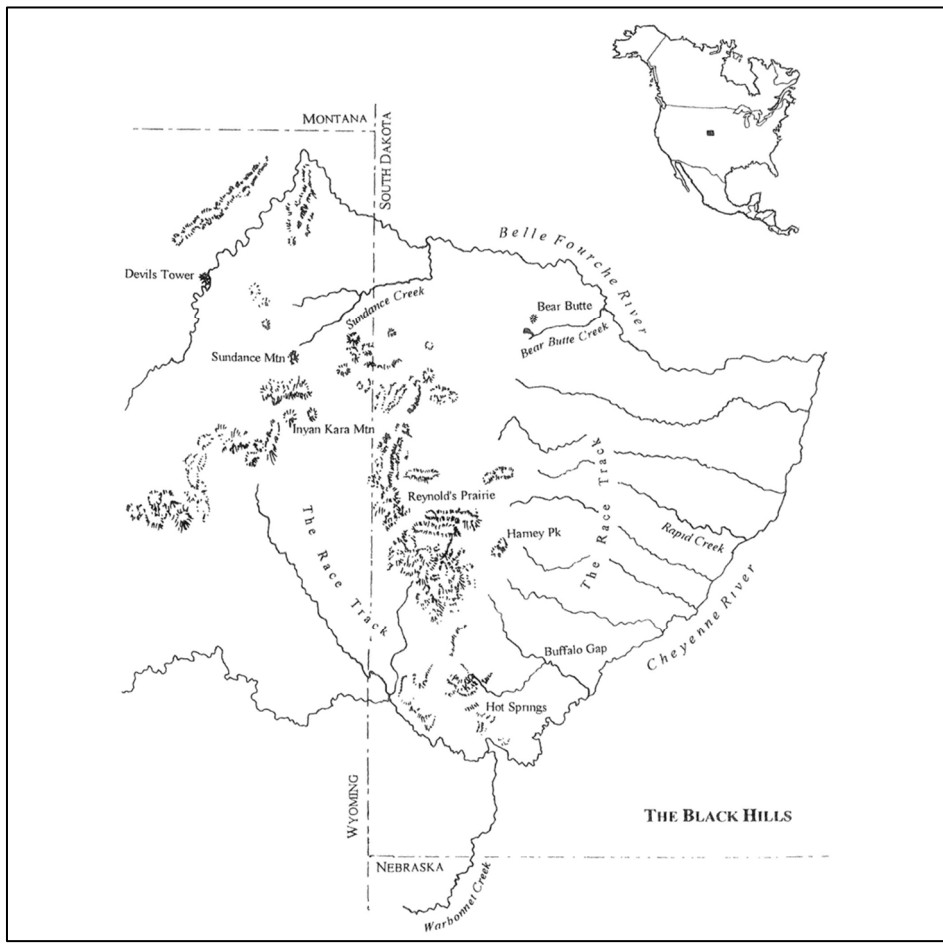

**Figure 4.** Map of Black Hills sacred geoscape [3].

Mateo Tepe is known to Native Americans from across North America and by Indigenous people from elsewhere in the world. It is the focus of many Native American prayers that ask for balance, healing, and understanding. Many traditional people who hold cultural interpretations of this geosite will personally visit it during sacred pilgrimages, prayers, and ceremonies in their lives.

Given the prominence of Devils Tower as a feature on the land, it is not surprising that the tower occupies a central position in several Native American tribes' Creation accounts. The tower's unique beauty, its prominence as a landmark during the days of early Euro-American exploration, and a need to create a sense of cultural nationalism led President Theodore Roosevelt to set it aside by Presidential Proclamation as the first U.S. national monument on 24 September 1906 [35,36].

The imposing physical quality of Devils Tower situated on a wide high plane qualified it as a symbol of U.S. national stability. With fluted stone columns, it is an imposing monolith of volcanic rock. The tower, which, from a distance, looks like a huge tree stump, rises above the plains and rolling hills in Wyoming. The surrounding scenery exhibits features of both the Black Hills and the Great Plains.

On U.S. Independence Day in 1893, local ranchers William Rogers and Willard Ripley drove stakes into one of the laccolith cracks, enabling them to climb to the summit. The ranchers took advantage of the national symbolism associated with Independence Day and became entrepreneurs; they reportedly collected USD 300 from a crowd that gathered to watch. Rogers unfurled a large U.S. flag on the summit [37]. On 4 July 1895, the ranchers varied the program by having Rogers's wife make the climb [37].

## 3. Results Heritage Geoparks and Geosites

Areas with special geology and topography are celebrated, preserved, and interpreted as geoparks andgeosites, which are largely distinguished by spatial scale. These are defined today as heritage parks and places because they are important in the collective lives of a people and are thus a central component of their heritage and contemporary culture.

Geoparks are recognized at the community, national, and international levels of heritage laws and regulations. The operational definition of geopark used in this analysis is virtually synonymous with that of the UNESCO definition of geopark, the scope and definition of which are managed under the voluntary authority of UNESCO's International Geoscience and Geoparks Program [38]. UNESCO provides a standard for geoparks and a certification service to territories that apply for formal status. Each nation-state in turn provides its own definitions and management recommendations. Cultural heritage studies are often a foundation of the establishment, interpretation, and management of geoparks and geosites [17,18].

The President of the U.S.A. Theodore Roosevelt proclaimed in 1906 that Devils Tower will be a monument (contemporarily referred to as a geopark) when he wrote:

*And, whereas, the lofty and isolated rock in the State of Wyoming, known as the "Devils Tower", situated upon the public lands owned and controlled by the United States is such an extraordinary example of the effect of erosion in the higher mountains as to be a natural wonder and an object of historic and great scientific interest and it appears that the public good would be promoted by reserving this tower as a National Monument with as much land as may be necessary for the proper protection thereof (Proclamation 658).*

## 4. Cultural Background

This analysis is based on the research and personal experiences of the authors who conducted funded Native American consultation research projects for U.S. federal agencies including the U.S. Air Force, the U.S. Fish and Wildlife Service, the U.S. Forest Service, and the U.S. National Park Service. Especially important resources to this analysis are a Native American Graves Protection and Repatriation Act (NAGPRA) study conducted in 1994 and personal communications between Stoffle and Vine Deloria regarding his expert witness

testimony for the legal case discussed later [38]. Other sources used in this analysis are published documents about the topic.

*4.1. Devils Tower NAGPRA 1994 Study*

The Native American Graves Protection and Repatriation Act (NAGPRA) study was commissioned by the NPS Applied Ethnography Program in Washington, D.C., (1) to identify individuals and tribes affiliated with the objects of cultural patrimony, sacred objects, or unassociated funerary objects at five NPS units, (2) to review those unit summaries, assist park or center staff in initiating consultation regarding those objects, and (3) to conduct a case demonstration of tribal consultation using the example of Pipe Spring National Monument.

Portions of the NAGPRA study are summarized here; however, the report [39] provides 28 pages of text, tables, and photos to convey study finding. These NAGPRA findings are selectively summarized here to the extent that they are useful for understanding this geosite.

Mateo Tepe figures in the traditions of some of the Northern Plains peoples, and it is for some the major goal of religious pilgrimage. The NAGPRA 1994 study identified ten tribes who are clearly culturally associated with Mateo Tepe. These are listed below with some references:

1. Arapaho
2. Assiniboine
3. Blackfeet, including (a) Pikuni or Piegan, (b) Kainah or Blood, and (c) Siksika or Blackfoot [40]
4. Cheyenne
5. Crow
6. Gros Ventre
7. Kiowa
8. Kiowa-Apache
9. Sarsi
10. Teton Dakota [41]

The following are a few of the memories of origin tribal people that were recorded in the NAGPRA 1994 study.

4.1.1. Cheyenne

Cheyenne speakers refer to the Devils Tower as Nakeove, which glosses as Bear Peak. The Cheyenne share essentially the same legend accounting for the creation of the striking stone formation with other northern Plains peoples. Children played Nakonistoz, or "bear play" [42]. Schlesier claims that the game's rules derived from proto-Tsistsistas (Cheyenne) bear-hunting rules [42].

4.1.2. Cheyenne Pilgrimages

Like other northern Plains peoples, Cheyenne people used pipes to petition persons believed to possess power to exercise it for the benefit of the group. One oral history related how adults gave four little boys four pipes to present to Stands in the Timber to persuade him to call bison during a period of hunger. Stands in the Timber smoked the four pipes in sequence, passing each around for others to smoke until the tobacco in it was consumed. The next morning, hunters found numerous bison [43].

Other text provides details about contemporary pilgrimages. This text is regarding Bear Butte but is typical of spiritual journeys to Mateo Tepe. When two Cheyenne men vowed that they would fast on the summit of Nowah'wus (Bear Butte in Black Hills, Dakota) in 1961, they carried the pipe to Willis Medicine Bull who agreed to instruct and lead them. When the pilgrims halted part way up the butte, Medicine Bull filled the pipe and offered it [44]. He also instructed the two fasting pilgrims to fill the pipe and offer it to Maheo and

to Sweet Medicine. Maheo is a spiritual being and possibly the source of the name Mateo Tepe which would mean his home.

### 4.1.3. Prayer Cloths

Cheyenne people leave offering cloths at sacred shrines, especially on top of Nowah'wus, the Sacred Mountain (Bear Butte, South Dakota). Another spelling is Noaha -vos, and another gloss is "The Hill Where the People Are Taught" [45]. In Tsistsistas (Cheyenne) cosmology, mountain peaks are especially sacred places because there, the deep earth and the near sky space come into direct contact [42]. The Rev. Peter J. Powell [44] has documented Cheyenne quests for power at Nowah'wus from the final days of United States military conquest of the Cheyenne people until the last half of the twentieth century. Those Cheyenne people about to undertake the arduous forced march south under military escort in 1878 left offering clothes tied to branches of trees growing on the summit of the peak [44].

When four elderly Cheyenne men made a pilgrimage to Nowah'wus in 1939, one left an offering cloth on the summit [44]. The eldest pilgrim had captured George A. Custer's guidon in 1876 on the Little Bighorn Battlefield and lived to become a healer.

In September 1945, the sacred Cheyenne Arrow bundle was opened at the base of Nowah'wus, and the Mahuts exposed "upon a bed of sacred white sage and offering clothes" [44]. Four pilgrims fasted halfway up the slope. One dreamed that a rider charged past him and down the side of the butte without any damage to man or mount [44]. Cheyenne men still sought visions, in other words, on their Sacred Mountain, illustrating the continuity of ethnic belief and behavior. The pilgrims to Sacred Mountain in the fall of 1950 carried approximately 75 offering clothes with them, gifts from Cheyenne people unable to make the pilgrimage. The pilgrims tied the offering clothes to shrubs and trees atop the Sacred Mountain [44].

In view of the Cheyenne cosmological perception of the relationship between supernaturally powerful sky and peaks, it is not surprising that offerings or prayer cloths are tied to trees near the base of Nakeove (Bear Peak or Devils Tower). Representatives' prayer clothes were observed during the NAGPRA 1994 study.

### 4.1.4. Kiowa

One Kiowa mythic legend accounts for the origin of the Black Hills, and another deals with the noted Bear Lodge (also known as Devils Tower or Tsó-ai—"tree rock", i.e., monument rock—near Sun Dance, Wyoming, which they claim is within their old country [35]. Ethnographer J. P. Harrington [35] was struck by the fact that Kiowas in Oklahoma preserved oral traditions of their ancestral residence in the Black Hills and, before that, on the head of the Missouri River and a legend accounting for both the creation of Ts'ou'a'e or Bear Butte and the Pleiades Constellation. The traditional Kiowa homeland was 650 miles distant and not seen by any living Kiowa or by the grandfather of any living Kiowa and yet vividly remembered in name and legend. The Kiowa legend attributed the northwestward leaning of the butte to Bear Woman's attacking and jumping against it from the southeast. The Kiowa applied the single name Ts'ou'a'e to Bear Butte and a different term to less spectacular buttes [35].

### 4.1.5. Teton Dakota

The Dakota legend concerning Mateo Tepe explains the origin of the butte. Three bears attacked three maidens picking wildflowers. The women climbed a rock seeking safety, but the sharp-clawed bears also climbed it. Seeing the women's predicament, the deity had the rock grow higher and higher. Finally, the exhausted bears fell hundreds of feet to their deaths at the base of the butte. The tough maidens fashioned a rope from their flowers and lowered themselves safely to the ground below [46]. The grizzly bears' claws scratched the butte into its striking and characteristic basaltic columns. Logically, the Dakota labeled the butte MateoTepe, or "Grizzly Bear's Lodge" [47]. According to Harrington[35], the

Lakota language had three additional terms for Devils Tower: (1) Witchátchepaha, or penis mountain; (2) Hinyánkaghapaha, or mythic owl mountain; and (3) Wanághipaha, or ghost mountain.

## 5. Tourism at Devils Tower

Scholars of tourism [48] have argued that touring can be understood as a sacred journey [49] consisting of (1) a normal home base and lifestyle; (2) a long, stressful, and sometimes difficult travel journey; and (3) a charismatic and dangerous destination with unique topographical features [50]. Some research argues that certain kinds of tourists are secular pilgrims [51]. During the touring journey and while at the destination, the tourist transforms into a liminal state and in some ways becomes another and more desired kind of person [52]. Liminal states are achieved during touring because it removes the person from normal life and provides a sacred state in the sense of being exciting, renewing, and inherently self-fulfilling. The tourist journey is a segment of our lives over which we have maximum control [49]. Scholars postulate that ordinary spaces encountered during touring can be converted into sacred spaces and even be revered and protected [53,54].

### 5.1. Liminality of Tower

Understanding the relationships between people and their environment is fundamental to understanding the cultural logic involved in ritual performances like pilgrimage. V. Turner is credited with the concept of "communitas", wherein pilgrims during a liminal state bond together with other travelers and with destination places [55,56]. Pilgrimages are important to a society's cultural connections to the local environment, and the incorporation of pilgrimage into ritual sequences affirms the collective valuation of particular places and the social memories inscribed in the landscape [57,58].

When pilgrims develop communitas relationships with people and places, both are socially reconceptualized in ways fitting the liminal experience. Simply put, a new social construct will be formed that redlines people and places who have been together in a pilgrimage. This analysis of Devils Tower and Mateo Tepe hinges on an understanding of social construction shifts resulting in the tower, on one hand, being from Hell and connected with the Devil and a pathway to hell and, on the other hand, becoming perceived as a sacred pathway to heaven or the Great Mystery in the sky.

### 5.2. Fifty Years of History

The early visitors to Devils Tower were faced with a daunting journey, according to a NPS historian who summarized 50 years of touring experiences in 1955 [59] (Mattison 1955). Colonel Dodge [60] is generally credited with giving the tower its present name "Devils Tower" in his book entitled The Black Hills[60]. Dodge maintained that the name derived from the Indian name, which translated as "The Bad God's Tower". Newton, who published his work on the U. S. Geological Survey from 1875 survey in 1880, said that the name Bear Lodge (*Mateo Tepe*) appears on the earliest map of the region but that the Devils Tower name is now in common usage.

The origin story of the name Devils Tower deserves some consideration especially in light of the park name change debate discussed below. It is an unlikely name to have been provided by a Native American person given that all the regional tribes have different names that are expressed in their language. Also, there is no Devil God or Bad God, Nor is there a concept of Hell in regional Native American religions. The Mateo Tepee name is a widely shared place name as the Native American people believe that the tower is a path to the Creator and the Afterlife. Instead, it is worth considering that the tower's name and association with the Devil better represent Western epistemologies regarding this spiritual being and a world below the surface of the Earth where he lives. These Western views, heavily influenced by Christianity, are reflected in the naming of a number of geosites: (1) Devils Lake, Wisconsin, formed in a deep volcano; (2) Devils Anvil, Arizona, a prominent volcanic butte overlooking the Colorado River in the Grand Canyon; (3) Devils

Hole, Nevada a deep natural hole that opens to a fast-flowing underground river; and (4) Devils Postpile, California, which, with geologically similar volcanic hexagonal basalt columns to Devils Tower, was established as a National Monument in 1911 by U.S. President William H. Taft. Like Devils Tower, these other geosites were given the name Devil by either western U.S. government surveyors or local Christian faith settlers. The Tower and Postpile geosites were made national monuments within a few years of each other and given the name Devil by the government. Neither was named by Native American people.

Access to the Tower before 1930 was largely via an unmaintained dirt road that required travelers to ford the Belle Fourch River seven times on their journey. Although the Tower was difficult to reach, an annual Independence Day national celebration was held there starting in 1893; the first celebration there featured food, music, and the first climbing. It is estimated that 1000 people came from as far as 125 miles away. It was 1930 before a full time NPS employee was assigned to the Tower, but his residence was elsewhere. A picknick was held at the monument on 4 July 1916 and was attended by 500 people. In 1917, Congress built a 12-to-16-foot-wide dirt road to the monument. A bridge over the Belle Fourch River was bult in 1928. Although visitors lacked shelter or facilities, they drank from a natural spring emerging from the base of the Tower. Tourists had to camp near its base between 1921 and 1930 as annual visitor numbers ranged from 7000 to 14,720. After 1930, commerce, travel associations, newspapers, and periodicals gave the Tower wide publicity as one of the wonders of the world. In 1941, there were 32,951 visitors, and by 1954, there were 100,919 visitors, reflecting the national boom in car-based tourism. From this time on most U.S. citizens simply could tour anywhere, however they wanted [50].

The early tourist journeys to the tower were long and arduous, and the tent camping stay was uncomfortable and dangerous. Tourist motives for visiting, their experiences during the visit, and the long-term perception of this place are difficult issues to ascertain. Scholars of these touring issues, especially those focused on charismatic and pilgrimage places, suggest the argument that the Devils Tower had become a sacred place for visitors.

*5.3. Name Change Debate*

Public pressure from regional Native American tribes and some conservation associations pushed the NPS and Congress to consider changing the name of Devils Tower to Mateo Tepe or Bears Lodge. Formally these requests for a name change began in 2014 when petitions were submitted to the U.S. Board of Geographic Names by tribal governments.

Newspaper reporter Andrew Rossie [61] wrote an article for the Cowboy State Daily Star that centered the views of the Wyoming Senate President Ogden Driskill whose family has lived next to the national monument for nearly 150 years. Driskill says that he will continue to fight to keep the Devils Tower name. Some key features of Senator Driskill's views and Rossie's research are paraphrased from the article below.

- The name Devils Tower is not—and never was—offensive, so there is not enough of a case to rename it. "It absolutely in no way was intended to be derogatory or offensive to anybody", he said. "It was named with the best interpretation they could find at the time. The intent is where it's at in all of this".
- There are many American Indian legends surrounding the tower's creation and variations on its name.
- The name Devils Tower came from the journals of Henry Newton, a geologist and mapmaker traveling with a U.S. Army expedition exploring the Black Hills in 1875. According to that history, Newton was told that the spectacular geologic formation was "The Bad God's Tower" and that tribes avoided the landmark and the surrounding valley because of its ominous association.
- Devils Tower is perhaps Wyoming's most recognizable landmark that most visitors know as Devils Tower. Beyond semantics and cultural history, Driskill believes changing the name would directly impact the communities in northwest Wyoming.

- A name change would not stop tourism altogether, but it could make many people unfamiliar with a landmark they already know. They certainly would not know what Bear Lodge is. It would be tough for tourism.
- Regardless of the debates and disparities, Driskill sees every piece of history associated with Devils Tower as stories that can, and should, be told. There is not a problem with the name, but there is a solution in the venue. We need a new visitor center.

While there have been many others engaged in the Devils Tower name change debate, Driskill certainly outlined the key issues. He is from a founding family who has heritage rights. The original name was Devils Tower. A name change would economically harm the State of Wyoming and tourism in general. These stipulations are not debated here. Both they and this analysis speak for themselves.

The NPS issued a summary statement [62], which outlines the many legal petitions by Wyoming politicians. While none has been passed by the U.S. Congress, the next petition will not be taken up again until 2025.

*5.4. Climbing Debate*

Public pressure from regional tribes and some conservation associations pushed the Federal District Court of Wyoming to consider voluntary restricting rock climbing on Devils Tower during the month of June. The following quotes are taken from the legal opinion of the court case regarding the NPS climbing management plan regarding rock climbing on Devils Tower [63]. See also Allison Dussia's cultural background and legal summary of this case (Dussia 2000–2001) [64]. Key points are highlighted below:

Devils Tower is a National Monument, as well as the place of creation and religious practice for many American Indians. Devils Tower is referred to by many American Indians as "Mato Tipila". Rock climbers use the Tower for recreational and commercial climbing ascents. Over the past 30 years, rock climbing on Devils Tower has dramatically increased, affecting the environment and the spiritual life Native American people. Plaintiffs, however, argue against the name change. These plaintiffs include: (1) Bear Lodge Multiple Use Association, an organization whose stated objectives are to develop management goals for natural resources and to maintain public access to the sites; (2) Andy Petefish, the owner of a commercial guiding service at Devils Tower; and (3) three long-time recreational climbers of Devils Tower.

To address the various concerns, the NPS developed a Final Climbing Management Plan for Devils Tower National Monument (FCMP). In addition to providing educational and environmental programs, the FCMP asks that climbers voluntarily refrain from climbing during the month of June when American Indians engage in the Sun Dance and other ceremonies. The Secretary of the Interior (Secretary) approved the plan. Bear Lodge Municipal Use Association and other climbers (Climbers) challenged the Secretary's approval, arguing the FCMP violates the Establishment Clause. The district court found the FCMP balanced the competing interests and observed the Constitution and was therefore within Secretarial discretion in managing Devils Tower National Monument. Our jurisdiction arises pursuant to 28 U.S.C. §1291. We (the Court) believe the Climbers alleged no injury and, therefore, lacked standing to sue.

The Intervenors Cheyenne River Sioux Tribe and members of numerous other Indian tribes have long viewed Devils Tower as a sacred site of special religious and cultural significance. The NPS explains, "archaeological evidence has revealed that the ancestors to the Lakota people inhabited the Devils Tower area as far back as 1000 A.D., while ancestors to the Shoshone people inhabited the area in the 1500's". The historical use of Devils Tower and surrounding areas by Lakota people was acknowledged by the Fort Laramie Treaty of 1868. Amici Concerned Scholars state, "Devils Tower is central to [the Indians'] etiological explanation of the universe". The most sacred religious artifact of the Sioux people is the White Buffalo Calf Pipe, given to them by White Buffalo Calf Woman at the beginning of Creation when she emerged from Devils Tower. The Tower is prominent in other religiously

relevant traditional stories of the Sioux, as well as in the cosmology of numerous other northern plains tribes.

According to Intervenor Romanus Bear Stops, a traditional spiritual leader of the Cheyenne River Sioux Tribe, Devils Tower is also a pilgrimage site where important liturgical functions are performed. At Devils Tower, Native American people partake in Sun Dances and individual Vision Quests. The Sun Dance is a group ceremony of fasting and sacrifice, which leads to spiritual renewal of the individual and the group as a whole. Sun Dances are performed around the summer solstice. Vision Quests are intense periods of prayer, fasting, sweat lodge purification, and solitude designed to connect with the spiritual world and gain insight. Sun Dances and Vision Quests, as well as individualized prayer offerings and sweat lodge ceremonies, require solemnity and solitude [63].

## 6. Analysis

Ethnological analysis involves comparing individual ethnographic cases to determine whether a cultural pattern can be documented. The confidence in the findings from the Mateo Tepe ethnography is briefly situated here with cases of relationships between Native American peoples and volcanos. Table 1 presents a selection of 11 studies about Native Americans and volcanoes. All ethnographic studies were conducted by research teams directed by Stoffle at the University of Arizona, University of Michigan, or the University of Wisconsin-Parkside during the past 30 years. The ethnological analysis of this set of studies affords the opportunity to generalize Native American traditional cultural responses to volcanos and to do so with many of the scholars who conducted the field-based ethnographic interviews.

**Table 1.** Native American volcanic studies by name, location, tribes, and author.

| Volcano or Volcanic Feature | Location | Tribes & Pueblos | Publication Title | Year of Publication | Author(s), Journal or Technical Report |
|---|---|---|---|---|---|
| Petroglyph Volcanos, West Mesa | Albuquerque, New Mexico | Multiple Tribes and Pueblos | Petroglyph National Monument Rapid Ethnographic Assessment [65] | 1992 | Evans, Stoffle, and Pinal. Technical Report for NPS. |
| Vulcan's Anvil | Lava Falls, Colorado River, Grand Canyon | Southern Paiute | Piapaxa Uipi (Big River Canyon): Southern Paiute Ethnographic Resource Inventory and Assessment for Colorado River Corridor, Glen Canyon National Recreation Area, Utah and Arizona, and Grand Canyon National Park, Arizona [66] | 1994 | Stoffle et al. Technical Report for NPS. |
| Schrumham Peak, Buckboard Mesa | Nevada Test Site, Nevada | Multiple Tribes | Storied Rocks: American Indian Inventory Interpretation of Rock Art on the Nevada Test Site [67] | 1999 | Zedeño et al. Technical Report. Technical Report for USDE and Desert Research Institute. |
| Black Butte | Pahranagut Valley, Nevada | Southern Paiute | Cultural Landscapes of the Sheep and Pahranagat Mountain Ranges: An Ethnographic Assessment of American Indian Places and Resources in the Desert National Wildlife Range and the Pahranagat National Wildlife Refuge of Nevada [68] | 2002 | Stoffle, Toupal, and Zedeño. Technical Report for NPS. |
| Black Canyon | Colorado River, Lake Mead National Recreation Area | Multiple Tribes and Pueblos | Shifting Risks: Hoover Dam Bridge Impacts on American Indian Landscapes [69] | 2004 | Stoffle et al. |
| Milford Flats South | Utah | Multiple Tribes | Tribally Approved American Indian Ethnographic Analysis of the Proposed Milford Flats South Solar Energy Zone [70] | 2011 | Stoffle et al. Technical Report for SWCA Environmental Consultants. |

**Table 1.** *Cont.*

| Volcano or Volcanic Feature | Location | Tribes & Pueblos | Publication Title | Year of Publication | Author(s), Journal or Technical Report |
|---|---|---|---|---|---|
| Mt. Trumbull | Arizona | Southern Paiute | Little Springs Lava Flow Ethnographic Investigation | 2013 | Van Vlack & Stoffle et al. Technical Report for NPS |
| Milk Mountain | Utah | Paiute | Kavaicuwac: A Southern Paiute Pilgrimage in Southern Utah. | 2018 | Van Vlack. |
| Sunset Crater | Flagstaff, Arizona | Multiple Tribes and Pueblos | Talking with a Volcano: Native American Perspectives on the Eruption of Sunset Crater, Arizona [71] | 2022 | Stoffle and Van Vlack. |
| Little Springs | Arizona | Southern Paiute | Dancing with Lava: Indigenous Interactions with an Active Volcano in Arizona [72] | 2022 | Van Vlack. *Environmental Communication.* |
| Mount Loa | Hawaii | Native Hawaiians | Native Hawaiians Believe Volcanoes Are Alive and Should be Treated Like People, with Distinct Rights and Responsibilities [73]. | 2022 | Stoffle |

The following are a few of the generalizations that can be derived from the previous list of 11 ethnographic studies. Some of these stipulations have been analyzed elsewhere [71,74].

- Volcanoes are perceived to be alive; sentient; having agency, including talking; and understanding their place in nature (relations with the elements of Nature including people), as these were defined at Creation.
- Volcanoes expect humans to co-interact with culturally defined patterns of respect. When appropriate co-interaction occurs, volcanoes can use their power to provide songs, medicine, ceremony, and be a source of balance (healing) occurring at local, regional, and world scales.
- Volcanoes can become angry if insulted or placated when they become the center of ceremony and offerings. Ceremonies conducted at volcanoes require isolation, time, and fires.
- Offerings placed at volcanoes define an ongoing relationship between themselves and humans and so should not be removed. Offerings at volcanoes are legally defined under NAGPRA.
- Volcanoes are powerful and they become stronger when other powerful beings (hawks, hummingbirds) and plants (Indian tobacco, medicine plants) come to co-reside with them.
- Native American people are obligated by birthright responsibilities to protect volcanoes from damage and harm.

Generalization is always problematic given the diversity of human culture and its trans-evolution through time. These stipulations, however, are actionable because they have been made by official tribal representatives over three decades in formal research projects. All of the 11 studies listed above were reviewed and approved by both the participating tribes and pueblos and the federal agencies who manage the volcanoes and surrounding lands.

## 7. Discussion

Worldwide and throughout time, volcanoes have been culturally special places for humans [75]. While there is no single kind of human response to currently active and past volcanoes, there is a general tendency among traditional peoples who have lived nearby volcanoes for centuries to believe such areas contain powerful forces and spiritual beings that can serve the betterment of human society at local, regional, and world scales of effect.

Volcanoes with adjacent traditional human societies are found at Vesuvius in Italy; at Mauna Loa in the Hawaiian Islands [76]; at Mt. Baekdu on the border of China and North

Korea [12,77]; at Mt. Fuji in Japan; in the volcanic islands of the Philippines; throughout the Aleutians Islands [78]; on the island of Bali in Indonesia [79]; at Kilimanjaro in Tanzania; and near Mount Pelee in Martinique, West Indies. Each of the volcanoes on the Big Island in Hawaii is the focus of religious commemoration practices and shrines [80–82]. Mt. Baekdu, the sacred origin-site of Koreans, is the northernmost point of a 1500 km-long mountain range that ends at another sacred mountain in South Korea and represents the physical, ecological, cultural, and spiritual core of the people [12,77]. In Japan, Mt. Fuji has a sacred history that is recounted and responded to in both the Shinto and Buddhist religions [83]. In Bali, the Water Temple religion focuses on the volcano and a sacred lake in its active caldera where the prime temple is located [81]. On the volcanic island of Camaguin traditional people, including high chiefs, are buried in urns to be transformed into glowing sulfur in the caldera [84]. Traditional Bantu people living near Kilimanjaro used pecked volcanic stones in a ceremony to teach youth secret ceremonies [85]. Greeks worshiped at the volcanic vents, where they established temples, including one that housed the Oracle of Delphi [86].

Devils Tower is clearly a disputed area that now has interested parties such as local and state communities, tourist businesses, climbers, and of course, more than twenty Native American groups who are culturally affiliated with this heritage location. The case is not unusual; the lack of problem resolution for the NPS managers is typical of such disputes. For some, it is a big rock that attracts tourists, a landmark for economic gain. For others, it is a portal to other sacred dimensions, through which White Buffalo Calf Women emerged in order to give the sacred pipe to Native American people. The issues of dispute are founded on an epistemological divide about the reality and the challenges of environmental equity or justice [87]. Environmental Justice is used to refer to efforts facilitating the return of traditional people to or restoring their presence in aboriginal lands as either new owners or co-stewards [88,89]. Throughout the world such issues have emerged, and new generations will need to secure multicultural and equitable solutions.

The United Nations recognizes that humans have rights related to their heritage places whether these be artificially built (archaeology) or natural (geosites). When these rights involve geosites, they include (1) identifying the cultural meaning of the heritage resource, (2) defining its boundary in space and time, and (3) sharing in managing its sustainable use by themselves and others. Clarity regarding this issue, however, is often clouded by the reality, in which multiple cultural peoples share heritage ties to the same geosite.

The heritage ties of cultural groups to geosites are particularly problematic; for example, the human uses of volcanos, waterfalls, and mineral deposits rarely leave a distinctive ethnic footprint. Volcanoes provided energy during lava flows, but lava rocks made by shaman on hornitos are taken away and only recognized by their corn or pottery imprints [72]. Waterfalls may be an origin location for a Native American people, but the only record of this event occurs in songs, stories, and pilgrimages [90,91]. Mineral deposits, like a salt lake [92,93] or a hematite deposit in a cave, are only documented when the salt is used in oral traditions or when the red paint is used to make a cultural petroglyph on a cliff [94].

The location of the boundary of geosites differs for most cultural groups. Scientists especially wish to find boundaries so that they can produce clear and repeated analyses. Each cultural group will have criteria which exceed the central characteristics of the geosite. Thus, it will be layered with meanings, places, and times. A volcano to some is defined by the limit of its magma flow, whereas to a Native American person it can include a nearby mountain where the volcano spiritually derives from. Most boundaries are clearly within what might be defined as normal time and space, but many cultural groups around the world understand the existence of portals to other dimensions [20,21], which usually have their own times. Therefore, a geosite at the edge of the Grand Canyon might be sacred for the Navajo people because, in another time, a Spiritual Being rested at this location during Creation [91]. In fact, for most Native American peoples, distinctive geological places

will have ties to other dimensions and time frames that are not recognized by western epistemology and science.

Given the complexity of defining cultural meanings and even the boundaries in time and space, the sustainable management of geosites is challenging. Critical will be the notion of environmental justice, wherein management decisions by agencies charged with sustainable use of the geosite are reached with full consideration—that is, with the co-management or co-stewardship of culturally affiliated peoples. Tourists who visit geosites are an amorphous interest group that often cannot provide specific management guidelines and too often is spoken for by economic organizations who profit from touring activities. These are relatively new management ideals for agencies that must resolve different cultural requests. Too often, agencies resolve conflicting user demands by using the voice of western science as the only valid guide.

**Author Contributions:** The authors contributed equally to all aspects of the research analysis, writing and editing of this manuscript. All authors have read and agreed to the published version of the manuscript.

**Funding:** This research received no external funding.

**Data Availability Statement:** Data are contained within the article.

**Acknowledgments:** We wish to recognize Vine Deloria, Jr., who shared cultural perspectives on Native American sacred places and landscape including Devils Tower. Bea Medicine conveyed the need to really understand and respect Native American culture. Members of the National Park Service Applied Ethnography Program, especially Michael J. Evans, Alexa Roberts, and David Ruppert, have been visionaries in designing and funding studies of Native places and natural resources in geoparks.

**Conflicts of Interest:** The authors declare no conflicts of interest.

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
