# Peer review of "Mateo Tepe or Devils Tower: Native and Tourist Differences in Geosite Interpretations"

_land, doi:10.3390/land13030357_

Round 1

Reviewer 1 Report

Comments and Suggestions for Authors

General comment:

The report does a fine job of presenting the complex history of one of the World’s great “geosites” or “cultural features”.  To my knowledge, no current users (Natives Americans, Park Service, visitors, earth scientists) refer to the Devils Tower National Monument as a “geopark”.  I suggest the authors observe that usage and not simply use the term “geopark” to attempt to be in vogue with current geoheritage usage.  They confuse the discussion of the Mateo Tepe feature.

After the use of the term “geopark” in line 189-190, the authors do no use “geopark” again this report.  I am not sure why they need the term “geopark” at all in this report.  I suggest that the authors drop the term “geopark” from this report.

The format for the report I was sent to review is in PDF format which does not allow editing as I review.  This I have referred my comments by Line in the PDF version.   

 Line

15          Abstract.  Authors state:   Thus, Mateo Tepe became the centrepiece of a geopark.”

I do not think it is correct to say that MT became the centrepiece of a geopark.  In 1906 there was no concept of a “geopark”.  I suggest dropping this sentence.  See comment below for lines 189-190. 

42          suggest replacing “tower” with “feature”

172        correct spelling:  unfurled

189-190              The President of the U.S.A. Theodore Roosevelt proclaimed in 1906 that Devils Tower will be a geopark when he wrote: …

Let’s be very careful here to not reinterpret history.  Roosevelt did not proclaim in 1906 that DT “will be a geopark”.  This is misleading.  He proclaimed the tower as a “National Monument…”.  I urge the authors and editors to be careful here.  The term “geopark” dates from the late 1980s and 1990s.  As stated here the authors imply the term was in use before that time – in 1906.  This is misleading. 

I urge that the abstract (line 15) also implies an earlier use of the term “geopark”.  Mateo Tepe became (in 1906) the centrepiece of a National Monument, not a geopark.  Let’s be very careful in the use of this terminology.     

254        Check spelling:  Maheo or Mateo?

370        correct spelling: “bult” should be “built”.

482        Table 1.  A useful table.  I’d suggest that there be a link between the publications cited in this table and the References table (line 592).  I tried to cross-walk the publications cited or mentioned in Table 1 with the References but could not find which refences were being cited.  Add a column to Table 1 with the numerical refence citation.

575        correct spelling of “geosite”

References

I am pleased to see this list of 77 References.  References are critical evidence of the Scholarship used by the report authors. I often learn quite a bit about the topic at hand and the method(s) used by the authors to research their topic.  What I find a challenge in this report is figuring out what if any organization there is to the Reference list.  Clearly the References are listed numerically as they are cited in the report.  The References are not organized by usual methods such as alphabetical list by author’s last lame, or date of the reference.  This makes it difficult to read through the Reference list to get an overview of how References are organized and grouped. I understand that the authors are simply following the format guidance by the journal – in this case the journal Land. 

Reviewer 2 Report

Comments and Suggestions for Authors

The paper titled “Mateo Tepe or Devils Tower: Native and Tourist Differences in Geosite Interpretations” by Richard Stoffle, Kathleen Van Vlack, Heather Lim, and Alannah Bell is professionally written and holds a ton of valuable information about the history, politics, and cultural conflicts associated with Mateo Tepe or Devil’s Tower geosite.  I think the paper is great and publishable as is.

My main critique of the paper for the authors to consider before publishing is to rethink the examples used to contextualize their research within the broader field. Specifically, it may be beneficial to examine other case studies that have explored the conflict between tourism and indigenous populations at geosites that are also considered sacred by indigenous peoples. For instance, the conflict between tourists and indigenous peoples at Uluru rock (Ayer’s Rock) in Australia presents a compelling parallel to the situation at Mateo Tepe. By focusing on the key issue of conflict between tourism and sacred cultural landscapes at geosites, the authors can more effectively situate their research within the field of geoheritage.

Reviewer 3 Report

Comments and Suggestions for Authors

Dear Authors,

Thank you for allowing me to read this article. The subject of the research is a globally recognised object. The text takes the form of an interesting essay, but less of a scientific article. This caused me to be a little lost when reading it and looking for the scientific context. Perhaps other viewers of its content may have a similar feeling. This is due to the fact that the manuscript lacks a clearly defined aim of the paper and lacks a 'Research methods and data sources' section, as well as a 'Conclusions' section with a statement of possible limitations of the study and recommendations. The 'Results' section is written in a rather confusing way and it is not really clear what research result it is. The manuscript therefore needs a lot of revision before it can be published.

The general comments are as follows:

  1. Only the characteristics of the object under study are described in the ‘Summary’ section. It does not state what the aim of the research was, what methods the authors used to achieve this aim, or what conclusion they reached from the research.
  2. The manuscript is defectively structured as far as its parts accepted in 'Land' are concerned. The manuscript lacks clearly separated sections ‘Research methods and data sources’, ‘Conclusions’. In addition, I have the impression that after the introductory part the results are described in the section ‘3. Results Heritage Geoparks and Geosites’, but in the further part of the manuscript the authors again give the basis of the research, i.e. ‘4. Cultural Background’ and ‘5. Tourism at Devils Tower’. In this 5.section, showing the volume of tourist movement in historical terms only, the authors analyse, for example, the origin of the name Devils Tower and then refer to the debate on naming changes. In my opinion, both questions should be addressed earlier in the text, where the authors cite the origin of the names in the languages of the indigenous peoples, and this section should already focus on the tourist dimension of the site–also in contemporary terms. The question of tourist interest in the site should be included in a separate section of the manuscript as one of the bases for naming considerations, debates and then also forms of tourism related to this site.
  3. It is only from the first sentence in the ‘6. Analysis’ section that the aim of the research is (I guess) apparent and I guess (I guess) the steps taken to achieve it are briefly described. Then the sources of the data are given, so perhaps this is the 'Research methods and data sources' section? The results of the analysis are given in further paragraphs (under Table 1) in this part of the manuscript, so I guess these are 'Results'. This needs to be organised accordingly.
  4. Authors should also align the citation of literature throughout the text and the final References list with the editorial requirements of Land.

In order to improve the manuscript, it is imperative that you consider the following detailed comments:

  1. What is the main question addressed by the research? The 'Introduction' section does not clearly outline the purpose of the research undertaken/the research problem referred to in the manuscript. The research gap that the research carried out is intended to fill is not indicated.
  2. What parts do you consider original or relevant for the field? What specific gap in the field does the paper address? Relevant to the multidimensional significance of the site under study is conceptualised in the section ‘2.Sacred Space Background’, from which, however, I feel that ‘3. Results Heritage Geoparks and Geosites’ does not follow. This is because the research gap that the research carried out is intended to fill is not indicated. The methodology used to carry out the research is not even described. Furthermore, the grouping of phrases in the section title: ‘3. Results Heritage Geoparks and Geosites’ makes it unclear exactly which contents of the manuscript are the authors' research results and which are not.
  3. What does it add to the subject area compared with other published material? From the content of the manuscript, it is difficult to see which are the result of the authors' own research and which are not. In my opinion, there are too many citations of the literature, including self-citations. It is difficult to extract from this the research results referred to in this manuscript. If this manuscript is a review, this should be clearly written and then its content structured so that the reader is in no doubt.
  4. What specific improvements should the authors consider regarding the methodology? What further controls should be considered? There is a lack of a clearly separated and described 'Research methods and data sources' section in the manuscript, so it is difficult to see what specifically can be improved methodologically.
  5. Please describe how the conclusions are or are not consistent with the evidence and arguments presented. Please also indicate if all main questions posed were addressed and by which specific experiments. At the beginning of section '6. Discussion', it is indicated that volcanoes were culturally special places for humans and then examples from the world are cited. In my view, the first two paragraphs have the characteristics of an introduction to a research problem that is not identified in the paper. Similarly, the fourth paragraph (lines 544-550) has such characteristics. In contrast, the content of the paragraph in lines 559-571 seems to be a conclusion rather than a discussion. I have a similar feeling about the content of the final paragraph in this part of the manuscript.
  6. Are the references appropriate? Generally the references are correct, but for example the reference to the paper is missing: Harp, E.L., Lindsay, C.R. 2006 Stability of Leaning Column at Devils Tower National Monument, Wyoming: U.S. Geological Survey Open-file Report 2006-1130, whose authors address the issue of geology and the use of the rock as a climbing site, which is also referred to by the authors of the manuscript.

In addition, the References list is missing: 

Evans, Stoffle, and Pinal. Technical Report for NPS. 2013

Stoffle et al. Technical Report for NPS. 1994

Zedeño et al. Technical Report. Technical Report for USDE & Desert Research Institute. 1999

Stoffle, Toupal, and Zedeño. Technical Re-port for NPS. 2022

Stoffle et al. Facility Siting: Risk, Power and Identity in Land Use Planning. 2004

Stoffle et al. Technical Report for SWCA Environmental Consultants. 2011

Van Vlack & Stoffle et al. Technical Report for NPS. 2013

Van Vlack. The International Journal of Intangible Heritage. 2018

Van Vlack. Anthropological Perspectives on Environmental Communication. 2022

Stoffle. The Conversation US, Inc. 2022

7. Please include any additional comments on the tables and figures and quality of the data. Some figures need to be resized/scaled, the more so as they are maps, i.e. 'Figure 1. 1879 Survey Map of Black Hills with Bear Lodge (Mateo Tepe) Used to Denote What is 53 Now Devils Tower7.', 'Figure 4. Map of Black Hills Sacred Geoscape3.'.

In my opinion, the Devils Tower Quadrangle Wyoming-Crook Co. map is better for correctly locating the survey area.

Round 2

Reviewer 3 Report

Comments and Suggestions for Authors

Dear Authors,

Thank you for clarifying and referring to my doubts as to the issues concerning the structure of the manuscript, methodological nature, literature and others. It seems to me that once the introduction has been supplemented with issues of a methodological nature, the reader will not have the doubts that I had after reading this essay.